# Surgical Anatomy of the Liver—Significance in Ovarian Cancer Surgery

**DOI:** 10.3390/diagnostics13142371

**Published:** 2023-07-14

**Authors:** Stoyan Kostov, Ilker Selçuk, Rafał Watrowski, Svetla Dineva, Yavor Kornovski, Stanislav Slavchev, Yonka Ivanova, Deyan Dzhenkov, Angel Yordanov

**Affiliations:** 1Department of Gynecology, Hospital “Saint Anna”, Medical University—“Prof. Dr. Paraskev Stoyanov”, 9002 Varna, Bulgaria; drstoqn.kostov@gmail.com (S.K.); ykornovski@abv.bg (Y.K.); st_slavchev@abv.bg (S.S.); yonka.ivanova@abv.bg (Y.I.); 2Research Institute, Medical University Pleven, 5800 Pleven, Bulgaria; 3Department of Gynecologic Oncology, Ankara Bilkent City Hospital, Maternity Hospital, 06800 Ankara, Turkey; ilkerselcukmd@hotmail.com; 4Department of Obstetrics and Gynecology, Helios Hospital Müllheim, 79379 Müllheim, Germany; rafal.watrowski@gmx.at; 5Faculty Associate, Medical Center, University of Freiburg, 79106 Freiburg, Germany; 6Diagnostic Imaging Department, Medical University of Sofia, 1431 Sofia, Bulgaria; svetladineva7@gmail.com; 7National Cardiology Hospital, 1309 Sofia, Bulgaria; 8Department of General and Clinical Pathology, Forensic Medicine and Deontology, Faculty of Medicine, Medical University—“Prof. Dr. Paraskev Stoyanov”, 9002 Varna, Bulgaria; ddzhenkov@gmail.com; 9Department of Gynecologic Oncology, Medical University Pleven, 5800 Pleven, Bulgaria

**Keywords:** liver morphology, liver anatomy, ovarian cancer surgery, anatomical variations, liver ligaments, hepatic veins

## Abstract

Introduction: Ovarian cancer is the leading cause of death among all gynecological malignancies. Most patients present with an advanced stage of the disease. The routes of spread in ovarian cancer include peritoneal dissemination, direct invasion, and lymphatic or hematogenous spread, with peritoneal and lymphatic spread being the most common among them. The flow direction of the peritoneal fluid makes the right subphrenic space a target site for peritoneal metastases, and the most frequently affected anatomical area in advanced cases is the right upper quadrant. Complete cytoreduction with no macroscopically visible disease is the most important prognostic factor. Methods: We reviewed published clinical anatomy reports associated with surgery of the liver in cases of advanced ovarian cancer. Results: The disease could disseminate anatomical areas, where complex surgery is required—Morrison’s pouch, the liver surface, or porta hepatis. The aim of the present article is to emphasize and delineate the gross anatomy of the liver and its surgical application for oncogynecologists. Moreover, the association between the gross and microscopic anatomy of the liver is discussed. Additionally, the vascular supply and variations of the liver are clearly described. Conclusions: Oncogynecologists performing liver mobilization, diaphragmatic stripping, and porta hepatis dissection must have a thorough knowledge of liver anatomy, including morphology, variations, functional status, potential diagnostic imaging mistakes, and anatomical limits of dissection.

## 1. Introduction

Ovarian cancer is the leading cause of death among all gynecological malignancies. Most patients present with an advanced stage of the disease [1,2]. The routes of spread in ovarian cancer include peritoneal dissemination, direct invasion, and lymphatic or hematogenous spread, with peritoneal and lymphatic spread being the most common among them [3]. Complete cytoreduction with no macroscopically visible disease is the most important prognostic factor [2,3]. The flow direction of the peritoneal fluid makes the right subphrenic space a target site for peritoneal metastases, and the most frequently affected anatomical area in advanced cases is the right upper quadrant [4,5]. Involvement of the right diaphragm is observed in 40% to 70% of cases [1,2,3,4,5,6]. Nevertheless, the disease could also affect Morrison’s pouch, the liver surface, or porta hepatis [6]. Therefore, oncogynecologists should master the upper abdominal anatomy before performing such critical surgical maneuvers—liver mobilization followed by diaphragmatic peritonectomy, dissection at the Morrison’s pouch or porta hepatis area. Bulky tumor localization to the liver is frequent and one of the most challenging parts of dissection during advanced ovarian cancer surgery [7]. Crucially, surgery in the liver region requires the harmonization of anatomical knowledge with surgical skills. Hepatobiliary surgeons generally deal with intrahepatic parenchymal disease, whereas oncogynecologists dissect the extrahepatic area, the peritoneum, and peritoneally attached tissues. The aim of the present article is to emphasize and delineate the gross anatomy of the liver and its surgical application for oncogynecologists. Moreover, the association between the gross and microscopic anatomy of the liver is also highlighted. Additionally, the vascular supply and variations of the liver are discussed.

## 2. Anatomical Lobes of the Liver

The liver is the largest gland in the body and is located in the right upper quadrant beneath the right hemidiaphragm [8,9]. According to morphological anatomy and from the anterior surface, the liver has two lobes, a larger right one and a smaller left one [9,10,11,12,13]. On the posterior surface, the liver has two more lobes—caudate and quadrate—which are part of the right lobe [14]. The two lobes of the liver are divided anteriorly by the falciform ligament, posteriorly by the fissure of ligamentum venosum, and inferiorly by the fissure of ligamentum teres. According to anatomists and most authors, the true anatomical demarcation line between the two lobes is the falciform ligament [9,10,12,13]. However, the functional demarcation line of the liver differs from the anatomical one (discussed later).

## 3. Morphological Variations of the Liver

### 3.1. Liver Shape Variations

The normal liver is wedge-shaped, with the narrow end pointing to the left. It resembles a five-sided pyramid [14,15,16,17]. However, the liver can have globular, quadrilateral, rectangular, square, conical, boot, or saddle shapes [14,15,16,17]. Srimani and Saha investigated 110 isolated formalin-fixed livers from adult cadavers. The authors found a normal wedge liver shape in 57.3% of the specimens. The other 42.7% of the livers had shape variations, of which quadrilateral (14.5%) and transverse saddle shapes (9.1%) were more common [16].

#### Significance of Liver Shape Variations in Ovarian Cancer Surgery

Radiologists and oncogynecologists should be familiar with different liver shape variations to avoid diagnostic errors and unwanted intraoperative surgical complications. Surgeons should be aware of variations in liver shape as in such cases, the vasculature or gallbladder may have a variant anatomical location (Figure 1).

### 3.2. Congenital and Acquired Variations of the Liver

Congenital liver variations commonly include accessory lobes, agenesis, or atrophy of lobes. In contrast, acquired variations develop over the course of an individual’s lifetime. They result from diaphragmatic, peritoneal, or other organ pressures over the liver [15]. The most widely used classification of morphological variations of the liver is the Netter’s classification (Figure 2) [18,19].

There are few studies on morphological variations of the liver as anatomists and hepatobiliary surgeons are more interested in the branching pattern of the hepatobiliary system and the vasculature of the liver [13,14,15,16,17,18,19]. Gross variations of the liver include diaphragmatic grooves, accessory fissures, agenesis or accessory lobes, and pons hepatis [13,14,15,16,17,18,19]. The most important gross variations of the liver associated with ovarian cancer surgery are diaphragmatic grooves, accessory fissures, Riedel’s lobe, hypertrophied papillary process (Spiegel’s lobe) or caudate process, and pons hepatis.

#### 3.2.1. Accessory Liver Fissures

The presence of accessory liver fissures in different lobes of the liver is the most common morphological variation. It should be stated that such fissures could be observed in all lobes of the liver. However, they are commonly found on the visceral surface of the liver. Additionally, they can be single or multiple [13,14,15,16,17]. Singh and Rabi [15] observed 70 formalin-fixed livers and found the presence of accessory fissures in 81.4% of all examined livers. Vinnakota and Jayasree investigated 58 livers and found an incidence of accessory liver fissures in 53.44% [17].

#### 3.2.2. Deep Diaphragmatic Grooves

Prominent vertical diaphragmatic grooves are most often found on the anterosuperior surface of the right lobe of the liver and rarely on the left. They can be single or multiple, ranging from two to six. Their depth ranges from 1 cm to 2 cm and the corresponding serosa is intact [14,15,16,20]. There is a difference between the sexes as these grooves are observed more frequently in women than men [20]. The presence of diaphragmatic grooves varies from 6% to 11.43% [14,15,16]. However, Macchi et al. [20] examined 48 human livers and found an incidence of these grooves of 40%. The deep diaphragmatic grooves are considered to be acquired as they are believed to result from costal pressure and invagination of the diaphragm muscle into the liver [14,15,16]. Mancchi et al. [20], however, concluded that these grooves could be a suitable marker for the portal fissures and for the superficial projection of the hepatic veins with their tributaries.

Deep diaphragmatic grooves and liver fissures are shown in Figure 3.

#### 3.2.3. Significance of Liver Fissures and Grooves in Ovarian Cancer Surgery

Liver fissures and grooves are of great clinical significance in ovarian cancer surgery as they represent potential sources of diagnostic imaging errors (particularly computed tomography (CT) imaging). Any fluid source in the grooves can mimic a cyst or metastatic tumor in the liver, an intrahepatic hematoma, or a liver abscess. Moreover, disseminated ovarian cancer cells in the diaphragmatic grooves or hepatic fissures could be mistaken for intrahepatic focal lesions [21]. Therefore, a thorough knowledge of the anatomy and variations of the liver surface may help to avoid unnecessary misdiagnosis.

Diaphragmatic grooves could be mistaken with Chilaiditi sign or syndrome, especially in cases of free air due to a perforation of abdominal organ [22].

Chilaiditi sign (CS) is defined as interposition of the colon (commonly transverse mesocolon) between the right liver lobe and the diaphragm. The sign represents a radiological finding of a gas between the right diaphragm and the right lobe of the liver [23,24,25,26,27].

The CS is known as Chilaiditi syndrome when it is accompanied by symptoms (pain, vomiting, constipation) and complications (intestinal obstruction, perforation, and ischemia) [22,23,24,25,26,27]. There are many theories for this untypical predisposition of the bowel—diaphragmatic (phrenic nerve palsy or congenital muscle loss), hepatic (weakness of the falciform ligament), abnormally long colon, ascites, and obesity [22,25,26,27]. There is also a theory which states that the diaphragmatic grooves are formed by a mesocolic tissue invasion of the adjacent anterior right liver lobe margins [23]. Yavuz et al. noticed the possible relation between diaphragmatic grooves, CS, and Chilaiditi syndrome. Therefore, the authors retrospectively investigated this possible connection on 2314 CT scans. The authors did not find statistical or significant correlation between diaphragmatic grooves and the syndrome. However, the authors concluded that the grooves are likely derived from the CS, as more than half of the patients with CS had diaphragmatic grooves (25 patients of 46 (54.3%) had grooves on the right liver lobe near the falciform ligament). Their theory is supported by the fact that the majority of grooves are found among the adult population. Nevertheless, Yavuz et al. mentioned that further studies are needed [23].

It should be stressed that Chilaiditi syndrome could be confused radiologically with diaphragmatic grooves. Cawich et al. reported a case of a patient with peptic ulcer perforation at the first part of the duodenum. The authors initially considered possible Chilaiditi syndrome as the patient had an air above the right lobe of the liver on preoperative radiographs. Intraoperative findings showed diaphragmatic grooves on the right liver lobe. Authors concluded that a true pneumoperitoneum with the presence of diaphragmatic grooves could be mistaken with Chilaiditi syndrome [22].

### 3.3. Riedel’s Lobe

Riedel’s lobe is defined as a downward tongue-like projection of the anterior edge of the right liver lobe. Riedel’s lobe is located right to the gallbladder. In the medical literature, it is also termed a floating lobe or “tongue-like” lobe. The incidence varies from 3.3% to 31% [28,29,30,31]. This anatomical variation is rather asymptomatic, although symptoms such as abdominal distension and torsion episodes are possible observations [28]. Riedel’s lobe is shown in Figure 4 and Figure 5.

#### Significance of Riedel’s Lobe in Ovarian Cancer Surgery

Riedel’s lobe can be confused with an enlarged lymph node or an unidentified abdominal mass on various imaging techniques. In cases of isolated metastatic lesions in Riedel’s lobe, resection is a possible option [13,28].

A few cases of primary malignant tumors or metastases to Riedel’s lobe have been described [32,33,34]: Soo et al. reported on a Riedel’s lobe metastasis from a ductal breast cancer [32]; Zamfir et al. observed a case of a 65-year-old woman with primary hepatocellular carcinoma arising from her Riedel’s lobe. The lobe was resected with “en-block” cholecystectomy [33]. Al-Handola et al. reported a case of 64-year-old woman with incidental observation of Riedel’s lobe and intrahepatic cholangiocarcinoma. The authors stated that there are unanswered associations between Riedel’s lobe and cancer. They concluded that the lobe could be considered a possible site for primary hepatocellular carcinoma or hidden metastases [34]. Notably, the majority of cases of Riedel’s lobe involvement by a malignant tumor affected the female population [32,33,34]. However, there is no reported case in medical literature of metastases to Riedel’s lobe by ovarian cancer. Perhaps there were such cases, but this liver pathology was probably neglected by oncogynecologists. Moreover, some authors believe that the lobe is a simple variant of liver anatomy, corresponding to hypertrophy of segments V and VI, rather than a true anatomical variation [35,36,37]. Additionally, the lobe can be a source of a living-related hepatic transplant [13,28]. Therefore, ovarian cancer metastases to Riedel’s lobe should be staged as FIGO stage IV, as this liver anomaly is actually part of the liver.

### 3.4. Papillary Process (Spiegel’s Lobe) and Caudate Process of the Caudate Lobe

The caudate lobe of the liver consists of three parts—the caudate process (lateral), the paracaval part, and the medial papillary process (medial). The papillary process is also known as Spiegel’s lobe. The medial papillary process can be hypertrophied, prominent, underdeveloped, and absent [12,13,14,15,16,38]. Singh and Rabi observed an enlarged and underdeveloped papillary process in 4.29% and 1.43% of 70 examined liver specimens [15]. Srimani and Saha found an enlarged papillary process in 21.8% of 110 livers examined [16]. The authors also observed the absence of the caudate and papillary processes in 16.4% of the investigated specimens [16]. Another study found a prominent papillary process in 32% of 90 formalin-fixed livers [39].

#### Significance of the Papillary Process (Spiegel’s Lobe) and Caudate Process of the Caudate Lobe of the Liver in Ovarian-Cancer Surgery

A normal or small papillary process of the caudate lobe can be mistaken for enlarged lymph nodes at porta hepatis on CT scans. In contrast, an elongated Spiegel’s lobe or caudate process might mimic a pancreatic body mass [38].

The hypertrophied Spiegel’s lobe of the liver is shown in Figure 6. The hypertrophied caudate lobe of the liver is shown in Figure 7. The elongated papillary process is shown in Figure 8.

### 3.5. Pons Hepatis

Pons hepatis, also known as pont hepatique, is defined as bridging over the ligamentum teres fissure between the quadrate and left lobes of the liver. The anatomical variations can be divided into three subtypes: I—no communication; II—membranous communication; III—a large parenchymal bridge, in which the left and quadrate lobes emerge as a connected lobe covering the umbilical vein [40,41,42]. The incidence of pons hepatis varies in the medical literature. Studies examining the morphological anatomy and variations of the liver reported different incidences of 22.86% [15], 30% [39], 35.5% [16]. Pons hepatis is shown in Figure 9.

#### Significance of Pons Hepatis in Ovarian Cancer Surgery

The pons hepatis is functionally insignificant but may appear as an extrahepatic mass (on CT scan or ultrasound) if metastatic ovarian cancer spreads here [16,43]. Moreover, in the case of pons hepatis, it may not be possible to observe the fissure of the ligamentum teres on a CT scan and the dimensions of the two main lobes of the liver (left and right) may be incorrect [39]. The pons hepatis, especially the zone where the ligamentum teres hepatis attaches to the liver, is a site of metastatic disease that can carry tumor implants.

## 4. Ligamentous Attachments of the Liver

It should be noted that most of the ligaments are located at the posterior (visceral) part of the liver. The falciform ligament originates from the umbilicus and runs into the anterior and superior surfaces of the liver. On the dome of the superior surface of the liver, the ligament merges with the Glisson’s capsule. The falciform ligament separates the superior and anterior aspect of the liver into two layers—the left layer continues medially, whereas the right runs laterally. After separation, the left and right layer of the falciforme ligament become the anterior portion of the coronary ligament [3,12,44]. The falciform ligament terminates inferiorly, where the ligamentum teres, also known as the round ligament of the liver, continues into a fissure on the inferior and posterior liver surface. The round ligament of the liver represents the remnant of the left umbilical vein of the fetus. The fissure is also known as fissure for ligamentum teres or umbilical fissure [3,12,44,45]. The latter is limited medially by the left hepatic lobe and laterally by the quadrate lobe of the liver [12]. The ligamentum teres is in continuity with the ligamentum venosum just before joining the left branch of the portal vein. The ligamentum venosum is a fibrous remnant of the fetal ductus venosus. It is located in a fissure, which is limited medially by the left lobe and laterally by the caudate lobe. The ligament is also known as the Arantius’ ligament. The ligament travels from the left branch of the portal vein to the IVC [46]. This connection serves to suspend the Arantius’ ligament in order to achieve good exposure of the left portal pedicle [46]. The gastrohepatic ligament, which is part of the lesser omentum, attaches superiorly to the fissure of ligamentum venosum [12,44,45,46]. The inferior vena cava ligament is a fibrous membrane which envelopes the IVC and continues posteriorly towards the lumbar vertebrae. It runs from the caudate lobe to the edge of the VII liver segment. It is also known as the hepatocaval or Makuuchi ligament [11,47,48,49,50]. Ligament dissection reveals the terminal extrahepatic part of the right hepatic vein and the groove between the right and middle hepatic vein [37,38,39,40]. It should be noted that the Makuuchi ligament consists of the components of the portal triad (portal vein, hepatic artery, and bile duct) and ectopic hepatocytes. Therefore, the ligament should be carefully ligated after dissection to avoid perioperative bleeding or bile leakage. Moreover, the ligament should be evaluated on preoperative imaging as it may have carcinogenic potential [49]. The hepatoduodenal ligament is a right-sided margin of the lesser omentum which connects the porta hepatis of the liver and the superior duodenal flexure. The hepatoduodenal ligament includes the visceral and parietal peritoneum and holds the portal triad. It also forms the anterior wall of Winslow’s foramen (epiploic or omental foramen). The epiploic foramen is a communication between the abdominal cavity and the omental bursa. The posterior wall of the hepatoduodenal ligament covers the subhepatic part of the IVC and passes over the caudate process of the liver [12,19].

The coronary ligament is the largest liver ligament. The ligament is inferred by the reflection of the diaphragmatic peritoneum onto the anterior and posterior surface of the right lobe of the liver. Furthermore, the coronary ligament separates into an anterior and a superior layer. The posterior layer courses caudally and attaches the liver to the right kidney, right adrenal gland, and posterior abdominal wall. The hepatosuprarenal ligament is formed by the attachment of the posterior layer of the coronary ligament to the right adrenal gland. The anterior layer of the coronary ligament connects the superior surface of the liver to the inferior surface of the diaphragm. There is a peritoneum-free zone in the diaphragmatic surface of the liver between the two coronary bands in the right lobe of the liver. It is called “bare area” or “area nuda”. Actually, there is no true coronary ligament on the left liver lobe (the layers of the coronary ligament likely to form the left triangular ligament) [51]. Therefore, terms such as “right coronary ligament” and “left coronary ligament” are incorrect. The left triangular ligament is longer than the right and consists of two parts—the anterior and the posterior. The anterior part merges superomedially with the left part of the falciform ligament, while the posterior part merges inferolaterally with the lesser omentum. The right triangular ligament derives from the continuation of the anterior and posterior layers of the coronary ligament. It is located at the apex of the “area nuda” [3,12,19,44,51].

Most of the ligaments of the liver and their attachments are shown in Figure 10, Figure 11, Figure 12, Figure 13 and Figure 14.

## 5. Liver Segments

As mentioned above, the division of the liver can be described from two perspectives—morphological anatomy and functional anatomy. However, the functional anatomy of the liver has been a topic that has generated many debates over the centuries. For decades, it was believed that the true division of the liver parenchyma was the falciform ligament. However, Sir James Cantlie performed an autopsy on a patient with an atrophic right lobe and observed that the demarcation of the atrophy was lateral to the falciform ligament. Hence, he proposed that the line which runs from the fundus of the gallbladder to the suprahepatic IVC (or middle hepatic vein) is the true functional division between the two main lobes of the liver. The Cantlie’s line relates to functional rather than morphological anatomy as it is useful when performing hepatectomies [8,52].

The most widely adopted functional classification of the liver was proposed by French anatomist Couinaud. He divided the liver into two hemilivers, four sectors, and eight functionally unique segments. Each segment contains a branch of the portal triad (portal vein, hepatic artery, and bile duct), and each segment has its own independent vascular inflow, outflow, and biliary drainage [8,9,44,53]. The portal branches run within the segments, whereas the three major hepatic veins pass between the segments [3]. The middle hepatic vein (Couinaud used the Cantlie’s line) divides the liver into left and right hemilivers [44,54]. Each of the two hemilivers contains two sectors separated by the scissure, consisting of the right and left hepatic veins. The left hepatic vein divides the left hemiliver into lateral and medial sectors, while the right hepatic vein divides the right hemiliver into anterior and posterior segments. The segments are numbered clockwise. Segment II (superior) and III (inferior) are in the lateral sector. Segment VI is divided into two subsegments—IVa (superior) and IVb (inferior)—which are located in the medial sector. Segments VI and VII are only visible from the posterior view of the liver [44]. Segment V (inferior) and segment VIII (superior) are located in the right anterior sector, whereas segment VI (inferior) and segment VII (superior) are located in the right posterior sector [8,54]. The caudate lobe is individual and corresponds to segment I as it has a different vascular supply (receiving blood vessels and biliary branches from both the right and left lobes of the liver) which is unique and variable [11,54]. Additionally, the caudate lobe is the only segment where veins drain directly into the IVC [53,54]. Liver segments are shown in Figure 15.

Additional delineation of the liver segments and other functional classifications are beyond the scope of this review as they are more useful to hepatobiliary surgeons. However, oncogynecologists should be familiar with different liver segments of the liver since surgeons or radiologists used to describe the localization of liver pathologies (intrahepatic tumors or ovarian cancer metastases) by using the Couinaud classification. Furthermore, Rosati et al. proposed a new anatomical–surgical classification for ovarian cancer metastases around the liver consisting of five categories: superficial metastases involving Glisson’s capsule without parenchymal infiltration (type 1); carcinomatosis along the lines of reflection between the liver and hepatic ligaments (type 2); metastases on the surface and fossa of the gallbladder (type 3); metastases in the porta hepatis area, including potential neoplastic involvement from the peritoneal site (hepato-duodenal ligament and Rouviere’s sulcus) as well as portal triad lymph nodes (type 4); and finally, parenchymal metastases (type 5), further subdivided into “superficial”, infiltrating <1 cm in depth, and “intra-parenchymal”, traditionally classified according to the liver segment. The classification seems logical for future clinical applications in ovarian cancer surgery as it mentions what type of metastases require a hepatobiliary surgeon [7].

## 6. Vascular Anatomy of the Liver

### 6.1. Arterial Supply of the Liver

The liver receives a quarter of the entire cardiac output. The vessels of the liver are the common hepatic artery, the portal vein, and the hepatic veins [12,44].

The common hepatic artery arises from the celiac trunk separately or in a common trunk with the splenic and the left gastric artery. The left gastric artery lies cranially to the common hepatic artery, whereas the splenic artery is located slightly to the left of the common hepatic artery [19]. The latter runs on the superior part of the duodenum. The common hepatic artery divides into three branches—gastroduodenal, proper hepatic artery, and right gastric artery. The gastroduodenal artery is the first branch after the separation; it runs caudally, and supplies the pylorus, the upper part of the duodenum, and the pancreas. The right gastric artery also has a caudal direction and continues within the lesser omentum along the lesser stomach curvature. The proper hepatic artery arises immediately after giving off the gastroduodenal and right gastric arteries. The proper hepatic artery proceeds cranially between the two layers of the gastroduodenal ligament and runs anterior to the portal vein and medial to the common bile duct. At the level of the porta hepatis, it divides into the left and right hepatic arteries. The left branch runs vertically to the base of the fissure for ligamentum teres and supplies the caudate lobe and segment II-IV. The right branch runs posteriorly to the common hepatic duct in 80–90% of cases and divides into the right anterior (supplies segments V and VIII) and the right posterior (supplies segments VI and VII) branches [3,8,9,10,11,12,19,44,45]. In most cases, the cystic artery arises from the right hepatic artery [19]. The arterial supply to the liver is shown in Figure 16.

### 6.2. The Portal Vein

The portal vein is the main vessel of the portal venous system formed by the union of the superior mesenteric, splenic, and inferior mesenteric veins. In the majority of cases, the inferior mesenteric vein and the left gastric vein drain into the splenic vein. The splenic vein drains into the superior mesenteric vein anterior to the IVC and posterior to the pancreatic neck at the level of the second lumbar vertebra. Moreover, numerous small veins (cystic veins and venous tributaries of the right gastric and gastroduodenal veins) also join the portal vein. The portal vein has no valves and is about 8 cm long [8,9,10,11,12,54,55,56,57]. The portal vein enters the hepatoduodenal ligament and is located posterior to the common bile duct (slightly to the right of the portal vein) and the proper hepatic artery (slightly to the left of the portal vein). At the porta hepatis, the portal vein divides into left and right branches. The left branch of the portal vein has a more horizontal and longer extrahepatic course compared to the right one. It is often divided into two portions: the “intrahepatic” portion enters the umbilical fissure, while the “transverse” portion enters the hilus. The left branch of the portal vein supplies segments I, II, III, and IV. The right branch of the portal vein divides into anterior and posterior portions. The anterior portion supplies segments V and VIII, and the posterior one supplies segments VI and VII of the liver [12,54,55,56,57]. The gross anatomy of the portal vein is shown in Figure 17.

### 6.3. Porta Hepatis

The porta hepatis is a transverse fissure on the inferior surface of the liver. The porta hepatis is approximately 5 cm long and extends from the neck of the gallbladder to the fissure for ligamentum teres and Arantius ligament. It is located between the quadrate lobe in front and the caudate process behind, and the lesser omentum attaches to its margin. From posterior to anterior, the left and right portal veins and the left and right hepatic arteries enter the porta hepatis; on the other hand, the left and right hepatic ducts leave it (Figure 11) [12,19,58]. Moreover, the hepatic nervous plexus and lymph nodes are also found in the porta hepatis [58].

### 6.4. Hepatic Veins and IVC

The main outflow tract of the liver is through one channel—the intrahepatic veins. In most cases, there are three major hepatic veins—left, middle, and right. These veins drain into the suprahepatic IVC. The right hepatic vein is larger than the left and middle hepatic veins and has a short extrahepatic course. In most cases, the middle and left hepatic vein form a common trunk before draining into the IVC. However, both veins could drain separately into the IVC [9,10,11,44,45,57]. The right hepatic vein drains the venous blood from the liver area located lateral to the Cantlie’s line (segments V, VI, VII, VIII). The middle hepatic vein lies in the Cantlie’s line and drains venous blood from the lower part of the medial segment of the left lobe and the inferior part of the anterior segment of the right lobe (segments IV, V, VIII). The left hepatic vein drains the venous blood from the upper part of the medial segment and the complete lateral segment of the left lobe (segments II, III, IV) [9,10,11,44,45,57]. As mentioned above, the veins of the caudate lobe drain directly into the IVC. They are inconsistent as their number varies from one to five [55,57]. Additionally, there are small hepatic veins (5–20) of varying sizes that usually drain blood from the posterior surface of the right lobe directly into the IVC. These veins normally drain some small parts of segments VII and VIII [12,55]. The umbilical vein terminates either in the left hepatic vein or in the junction of the left and middle hepatic veins [11]. The IVC has a close relationship to the liver and can be divided into suprahepatic, intrahepatic and subhepatic IVC. It is located posterior to the pancreas, duodenum, porta hepatis and the posterior surface of the liver. The small hepatic veins drain into the IVC in the bare area of the liver [9,10,11,44]. The gallbladder and bile duct, however, are not the subject of this review. The relationship between the bile duct and the hepatic duct to the closest anatomical structures has been clearly described [8,10,11]. Hepatic veins are shown in Figure 18 and Figure 19.

## 7. Macroscopic and Microscopic Anatomy of the Liver—One Entity

For a better understanding of the vascular anatomy of the liver, surgeons should also be familiar with its microstructure. The microscopic anatomy of the liver lobules is described in the article. The liver lobules are sheets of connective tissue with a three-dimensional structure and a hexagonal shape. In these lobules, hepatocytes are arranged in cords, which are divided by sinusoids where blood passes without resistance. Each hepatic lobule contains three zones (periportal—close to the portal triad; mid- and pericentral—near the central vein) and two structures—a central vein and a portal triad. The latter is located in the periphery of the lobules and consists of a branch of the proper hepatic artery, a branch of the portal vein, and bile duct. Lymphatic vessels, a few inflammatory cells and the vagus nerve branch accompany each portal triad. The central vein is located in the center of the liver lobules and drains into the hepatic veins. Blood from the tiny branches of the hepatic artery and portal vein passes through sinusoids in a central direction and drains into the central vein. The bile ductules transport bile from the canals of Hering, opposite to the blood direction, and drains it in the liver ductile in the portal triad [12,59]. The microscopic anatomy of the liver lobule is shown in Figure 20.

## 8. Significance of Liver Anatomy in Ovarian Cancer Surgery

### 8.1. Liver Mobilization

Full liver mobilization is required in cases of moderate- to large-volume disease involving the peritoneum of the right diaphragm. This maneuver provides access to the right diaphragm [60]. The procedure requires the following steps [3,44,58,59,60,61]:Step 1: dissection and transection of the falciform ligament containing ligamentum teres in a cranial direction up to the level of its bifurcation into the coronary ligament (transection of the falciform ligament with an electrosurgical device is preferable);Step 2: dissection and division of the left triangular ligament;Step 3: dissection of the anterior and posterior layers of the coronary ligament on the left liver lobe;Step 4: dissection and division of the hepatogastric ligament;

The development of steps 2–4 allows the rotation of the liver on its vascular axis, thus avoiding the “hinge” effect between the immobile left lobe and the mobilized right;

Step 5: dividing (from medial to lateral) the anterior layer of the coronary ligament in the right liver lobe;Step 6: dividing the right triangular ligament;

The development of steps 5 and 6 allows medialization of the right hepatic lobe, exposing the right paracolic gutter and Morrison’s pouch;

Step 7: dissection and transection (from lateral to medial) of the posterior layer of the coronary ligament.

This step provides access to the bare area, the right kidney, and the right adrenal gland.

Step 8: dissection between the liver surface and the ventral aspect of the IVC.

It should be stated that there are many different surgical techniques for liver mobilization, depending on the surgeons’ preference and the extent of the tumor in the right upper abdomen. For instance, some surgeons use an extraperitoneal rather than a transperitoneal approach. Moreover, for bulky tumors on the coronary ligament in the right lobe, the surgeon may attempt to expose the bare area as the first step of dissection [1,2,3,4,5].

#### 8.1.1. Step 1. Tips, Tricks, and Attention during Dissection

During the dissection and transection of the falciform ligament, surgeons should pay attention to the presence of the paraumbilical veins—Sappey’s and Burrow’s veins [62,63,64,65,66]. Sappey’s veins can be divided into the superior and inferior group. The superior veins drain the cranial part of the falciform ligament and the median diaphragm. These veins terminate in the peripheral portal branches of the left lobe of the liver. The superior Sappey’s veins anastomose with the phrenic, internal thoracic, superior epigastric veins, and peripheral branches of the left portal vein [62,65].

The Inferior Sappey’s veins drain the caudal portion of the falciform ligament and anastomose with the inferior epigastric and subcutaneous veins. The inferior group of veins could drain into the left portal vein, into the umbilical vein (not common), or directly into the caudate liver lobe [62,63,64,65,66].

Burrow’s veins were first described by Burrow in 1838. He described a pair of veins which ascend from the inferior epigastric veins alongside the umbilical vein. These veins do not terminate directly in the liver as they drain into the umbilical vein [62,65,67]. Burrow’s veins anastomose with branches of the inferior epigastric vein and the inferior group of Sappey’s veins [65].

All of these veins participate in portocaval anastomoses and become a collateral pathway [62,63,64,65].

Sappey’s and Burrow’s veins are shown in Figure 21 [62,63,64,65].

In some cases, metastatic cancer at the base of the round ligament of the liver could be a reason for suboptimal debulking during ovarian cancer surgery [68,69,70,71]. Moreover, recurrences at the ligamentum teres have also been described [72]. The ligamentum teres fissure lies between the III and IVB liver segments. As mentioned above, pons hepatis is a bridge of liver parenchyma covering the round ligament and forming a tunnel covered by peritoneum. Therefore, this tunnel should be dissected in patients with advanced ovarian cancer and peritoneal carcinomatosis. However, in cases of semi-closed pons hepatis, it is difficult to expose the terminal part of the round hepatic ligament. Dissection of the complete pons hepatis facilitates identification of the base of the round ligament, as the tunnel is the continuation of the peritoneal tissue [67,68,69]. At the beginning of the dissection, the risk of portal triad injuries is relatively low. Particular attention should be paid to the preparation of the ligament base. There is a risk of injury to the left portal vein, the left hepatic artery, and the left bile duct (Figure 22 and Figure 23) [68,69,70,71].

#### 8.1.2. Step 2 and 3. Tips, Tricks, and Attention during Dissection

Care should be taken to avoid injury to the left hepatic (LHV) and middle hepatic vein (MHV). If there is no massive carcinomatosis in the diaphragmatic peritoneum, the surgeon may follow the left inferior phrenic vein. In most cases, this vein empties into the LHV [73]. Variations of the LHV and MHV are less common than those of the right hepatic vein. Sureka et al. examined hepatic vein variations with multidetector CT in 500 patients. The authors found that LHV and MHV shared a common trunk In 81% of cases [74]. The incidence of the common trunk before the confluence with the IVC is found in approximately 65–85% of the population. It is therefore a normal anatomical finding rather than a variation [74,75]. Venous variations such as accessory veins, the segment IV vein, the anterior superior segmental vein, or the umbilical vein, which drain into the LHV or MHV, are not the subject of the present article as these anomalous veins have intrahepatic course [74,75,76]. During this step, surgeons should be careful about the extrahepatic variations of LHV and MHV. One such variant is an LHV draining directly into the coronary sinus [77,78,79]. In such cases, the LHV is a single vessel that arises from the left liver lobe, crosses the diaphragm, and drains into the coronary sinus close to its orifice. It is often asymptomatic and associated with other cardiac or vascular anomalies (left or double superior vena cava). It is an extremely rare variant, as fewer than 10 cases have been described [77]. However, oncogynecologists should be familiar with this particular variation because anomalous LHV has a long extrahepatic course that differs from the course of suprahepatic IVC. It should also be mentioned that in rare cases the hepatic veins can terminate in the left atrium [80,81].

#### 8.1.3. Step 4. Tips, Tricks and Attention during Dissection

The key step in dissection of the hepatogastric ligament is through the pars flaccida. It is a region almost transparent over the caudate lobe of the liver [82,83]. During this step, surgeons should be aware of one particular vessel variation—the left hepatic artery arising from the left gastric artery. In such cases, the left hepatic artery passes through the midline of the lesser omentum. The incidence of this variation varies from 12% to 34% of the population [84]. Fortunately, ligation or transection of the anomalous artery is not associated with liver dysfunction. A transient increase in liver enzymes is often observed [84,85]. However, in rare cases, the common hepatic artery may originate from the left gastric artery. Injury to this anomalous artery is associated with liver ischemia and necrosis since all arterial supply to the liver is derived from the anomalous common hepatic artery [86].

#### 8.1.4. Steps 5 and 6. Tips, Tricks, and Attention during Dissection

Surgeons should be mindful of the right suprahepatic vein and suprahepatic IVC. The latter is located slightly to the left side of the anterior layer of the coronary ligament in the right liver lobe [87]. The distance between the anterior layer of the coronary ligament and the suprahepatic veins is approximately 1 cm [88]. Injury of the right hepatic vein (RHV) is more common than the LHV and MHV, as the RHV has a longer extrahepatic course. The safety plane of dissection is juxtaposed to the liver parenchyma [3]. On the right diaphragmatic surface, the right phrenic vein could be observed running toward the RHV. The right phrenic vein could be a landmark for identifying the RHV [44].

Over the years, authors have proposed different classifications of RHV variations. However, they are mostly associated with its intrahepatic course [74,89,90,91]. Nakamura and Tsuzuki classified the RHV variations into four types. Of greatest importance for ovarian cancer surgery is type IV in which two superior accessory hepatic veins are present. An accessory right anterosuperior hepatic vein drains into the RHV, and an accessory right posterosuperior hepatic vein, which has an extrahepatic course, drains into the IVC (superiorly to the RHV) [89,90,91]. The incidence of type IV variation varies between 4.2% and 11.7% [89,90,91]. Another anomalous vessel that could be injured during these steps is RHV, which drains into the IVC after passing through the caval aperture [44].

#### 8.1.5. Steps 7 and 8. Tips, Tricks, and Attention during Dissection

These surgical steps include dissection around the bare area of the liver in the immediate vicinity of the right adrenal gland which is located behind liver segment VI. Notably, segments V and VI are most commonly resected during ovarian cancer surgery [92].

The most common variation of the hepatic venous system is an accessory right inferior hepatic vein. The incidence of this anomalous vessel varies from 21% to 48% [90,91,93]. The accessory right inferior hepatic vein drains the inferior segments of the liver (mainly segments VI and VII) and may be larger and more dominant compared to the normal vein [93,94]. In 15% of cases, this vein runs ventrally to the hepatocaval ligament (Makuuchi ligament) and can be injured during step 8 [44]. An accessory right inferior hepatic vein is shown in Figure 24.

Safe dissection between the liver surface and the ventral aspect of the IVC requires awareness of the presence of small retrohepatic veins draining directly into the IVC. These veins are commonly observed in the right hepatic lobe of the liver [3]. In addition, there are additional caudate liver lobe veins, which also terminate in the IVC. These veins are known as Spigelian veins [95].

The bare area (area nuda) and additional hepatic veins, which drain into the IVC, are shown in Figure 25 and Figure 26. Step 6, 7 and 8 is shown in Figure 27.

### 8.2. Porta Hepatis and Hepatoduodenal Ligament Dissection

The dissemination of porta hepatis from ovarian cancer varies in medical literature, as authors report on both peritoneal and lymphatic tumor spread [96,97,98]. Raspagliesi found that 19% of patients with advanced ovarian cancer had peritoneum dissemination at porta hepatis [96]. Tozzi et al. reported that 14.3% of examined patients with advanced ovarian cancer had dissemination of porta hepatis and hepatoceliac lymph nodes. Eighteen patients out of thirty-one had only porta hepatis peritoneal involvement [97]. Donato et al. reported for 4.5% portal nodes metastases among 55 women with advanced ovarian cancer and hepatobiliary involvement [98].

There are different techniques for porta hepatis dissection. Some authors use vessel loop through the epiploic foramen to encircle the hepatoduodenal ligament (part of Pringle maneuver), whereas others carry out the Kocher maneuver to provide enough space for dissection. Tozzi et al. perform both maneuvers during dissection. The Kocher maneuver represents a medial pancreatoduodenal mobilization. The head of the pancreas and the first, second, and proximal third portions of the duodenum are mobilized. The procedure is performed easily as there is an avascular plane below the duodenum and the pancreatic head. The mobilization continues at the level where the left renal vein drains into the IVC (Figure 28) [96,97,98,99,100].

The Pringle maneuver represents a vessel loop which encircles the hepatoduodenal ligament and its structure (Figure 28). It is used to minimize blood loss during different types of hepatic resection [101].

The dissection starts with opening the anterior peritoneum of the hepatoduodenal ligament at a tumor free area. The proper hepatic artery and the common bile duct are identified. The vessels loop is retracted medially, and the posterior peritoneum of the ligament is dissected from the dorsal aspect of the portal vein. When all three anatomical structures of the hepatoduodenal ligament are identified and mobilized, the dissection continues in a retrograde fashion until the hepatic hilum. Enlarged lymph nodes are resected. Moreover, the peritoneal stripping of the ligament continues mediolaterally and posteriorly to complete the circumferential dissection. Surgeons should be aware of vessels variations (replaced right hepatic artery or left hepatic artery arising from the superior mesenteric artery) or biliary tree variations [96,97,98,99,100,102].

### 8.3. Gall Bladder Fossa Dissection

Ovarian tumor dissemination to the gall bladder fossa is often followed by a cholecystectomy. Rosati et al. mentioned that in such cases, a hepatobiliary surgeon is required [7]. The peritoneal tissue covering the gall bladder is a common anatomical area of dissemination. In such cases, surgeons dissect the peritoneal fold between the gall bladder and the duodenum in order to identify the disease. Gall bladder removal should be performed only in cases when optimal cytoreduction could be achieved [3,102].

## 9. Preventing Iatrogenic Injury during Liver Mobilization

Multidisciplinary management and preoperative collaboration between radiologists and oncogynecologists can help to avoid injury of anomalous vessels during liver mobilization. Preoperative detection of liver pathologies or variation can be performed by transabdominal ultrasonography (US), contrast-enhanced US, CT, MRI, multislice CT, diffusion-weighted MRI, or dynamic 3-phase contrast-enhanced MRI [103]. Intraoperative ultrasound is better suited for intrahepatic variations during surgery than for extrahepatic ones [104].

Awareness of vascular variations can also be helpful for interventional radiologists examining liver metastases, e.g., transarterial embolization (TAE), transarterial chemoembolization (TACE), or selective internal radiation therapy (SIRT). The latter methods are used somewhat more often in gynecological tumors of neuroendocrine origin. Knowledge of anatomical liver variations is also essential for radiofrequency ablation (RFA) [105,106].

## 10. Conclusions

Surgical procedures around the liver area are common in patients with advanced ovarian cancer and peritoneal carcinomatosis. Oncogynecologists perform liver mobilization, diaphragmatic stripping, and porta hepatis dissection. Therefore, oncogynecologists must have a thorough knowledge of liver anatomy, including morphology, variations, functional status, potential diagnostic imaging mistakes, and anatomical limits of dissection.

## Figures and Tables

**Figure 1 diagnostics-13-02371-f001:**
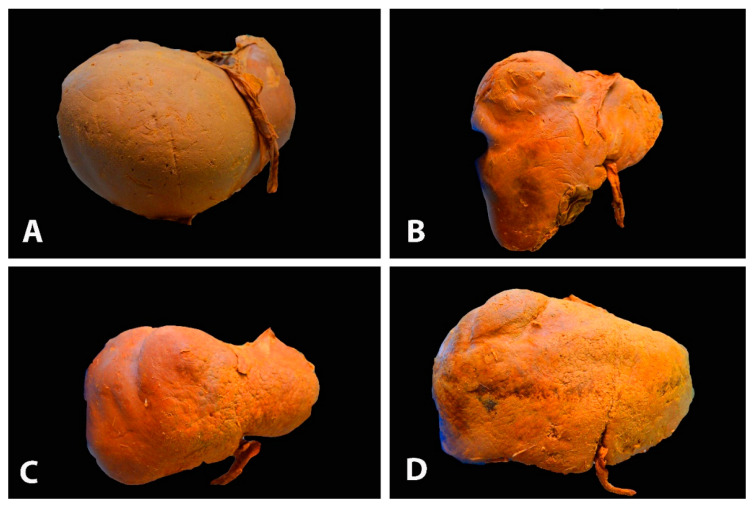
Different gross shapes of the liver—anterior surface of the liver (author’s own material): (**A**) globular shape; (**B**) conical shape of the right liver lobe and the small left lobe; (**C**) quadrilateral shape; (**D**) rectangle shape.

**Figure 2 diagnostics-13-02371-f002:**
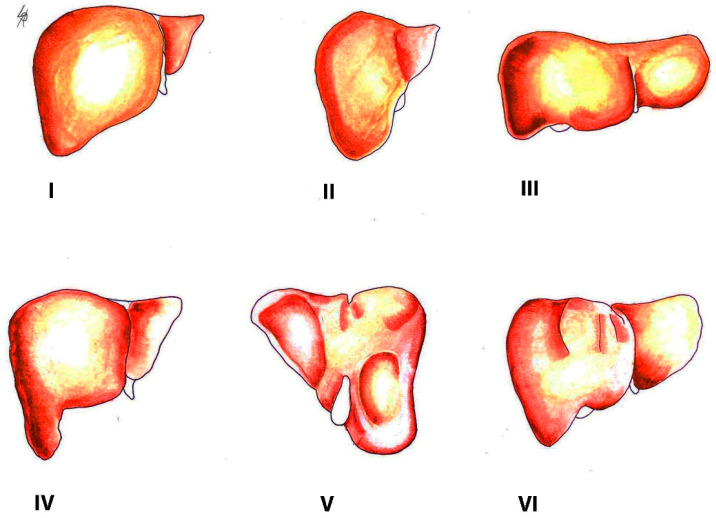
Morphological variations of the liver according to Netter’s classification (modified from references [18,19]). Type **I**—small left liver lobe and deep costal impression; Type **II**—left lobe atrophy; Type **III**—saddle-like liver with hypertrophied left lobe; Type **IV**—Riedel’s lobe; Type **V**—deep renal impression and corset construction; Type **VI**—diaphragmatic grooves.

**Figure 3 diagnostics-13-02371-f003:**
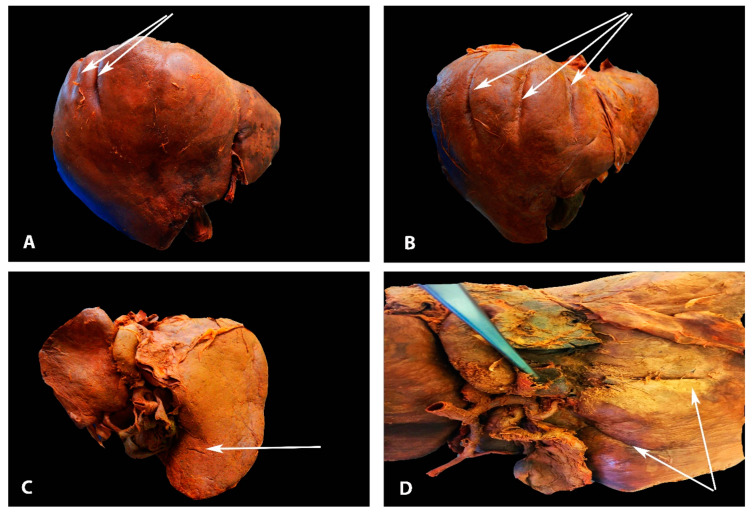
Deep diaphragmatic liver grooves and liver fissures (author’s own material). (**A**)The arrows show two diaphragmatic grooves on the right liver lobe (anterior liver surface). (**B**) The arrows show three diaphragmatic grooves on the right liver lobe (anterior liver surface). (**C**) The arrow shows liver fissure on the right lobe (posterior liver surface). (**D**) The arrows show two liver fissures on the right lobe (posterior liver surface).

**Figure 4 diagnostics-13-02371-f004:**
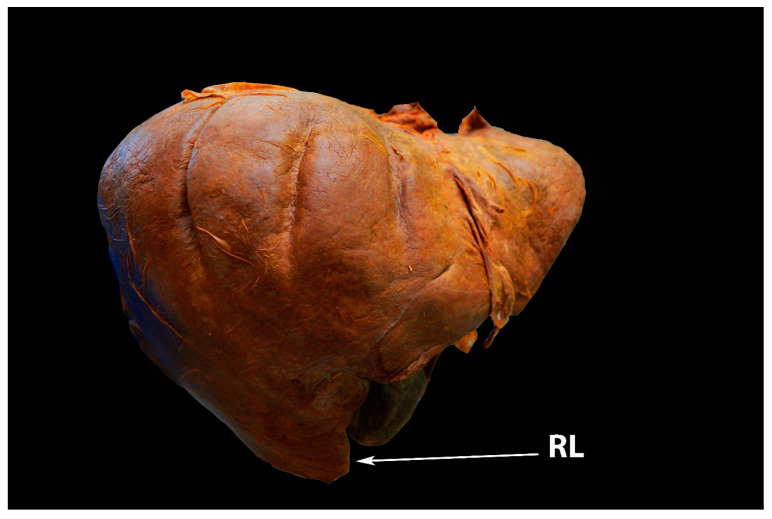
Riedel’s lobe of the liver—anterior liver surface (author’s own material). RL—Riedel’s lobe.

**Figure 5 diagnostics-13-02371-f005:**
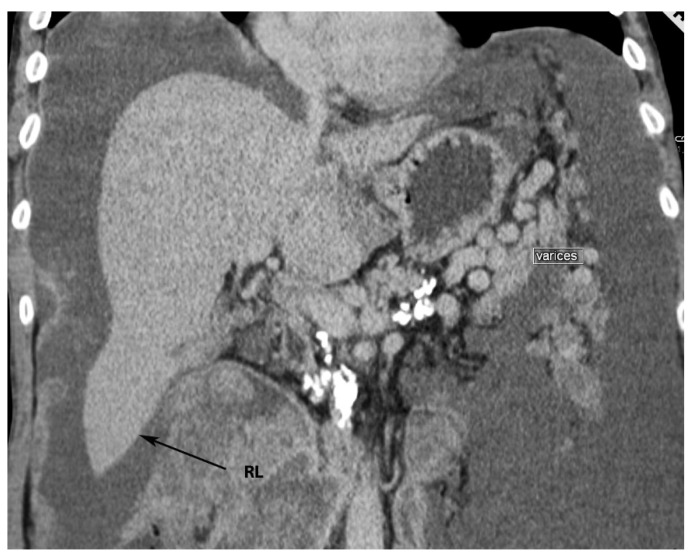
Riedel’s lobe on CT (author’s own material). RL—Riedel’s lobe.

**Figure 6 diagnostics-13-02371-f006:**
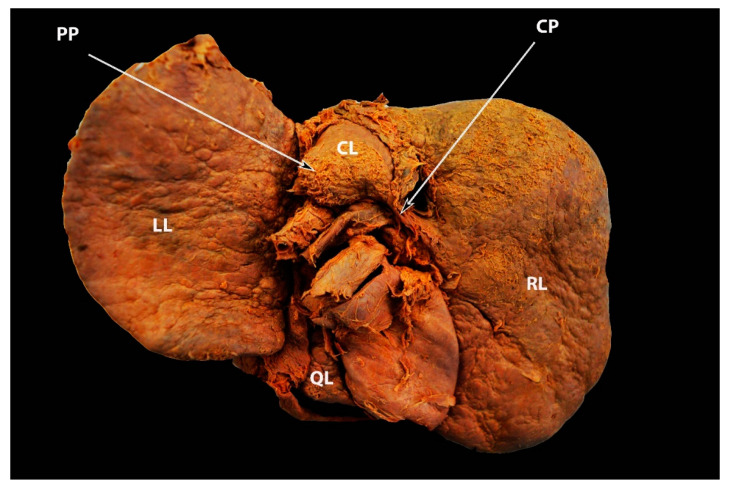
Elongated caudate process and hypertrophied papillary process showed by arrows—posterior liver surface (own material). LL—left lobe; RL—right lobe; QL—quadrate lobe; CL— caudate lobe; CP—caudate process; PP—papillary process.

**Figure 7 diagnostics-13-02371-f007:**
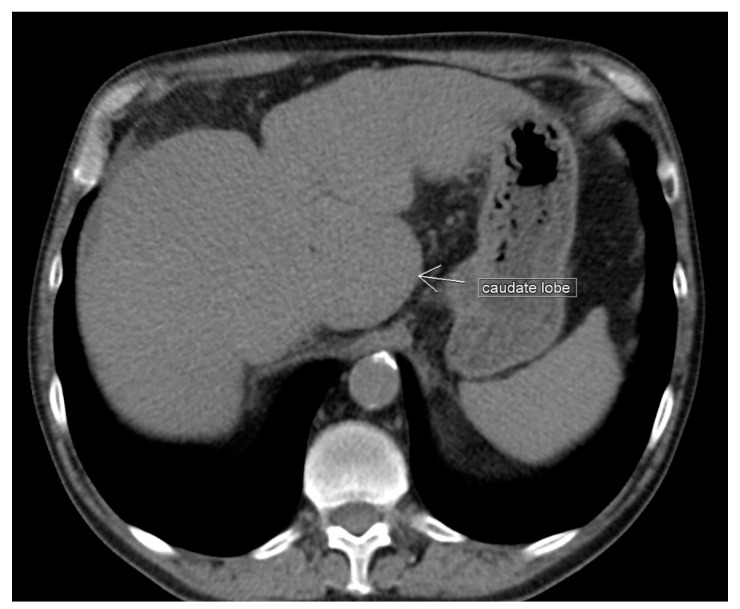
Unenhanced transverse abdominal CT liver level with isolated caudate lobe hypertrophy (own material). The arrow shows isolated caudate lobe hypertrophy.

**Figure 8 diagnostics-13-02371-f008:**
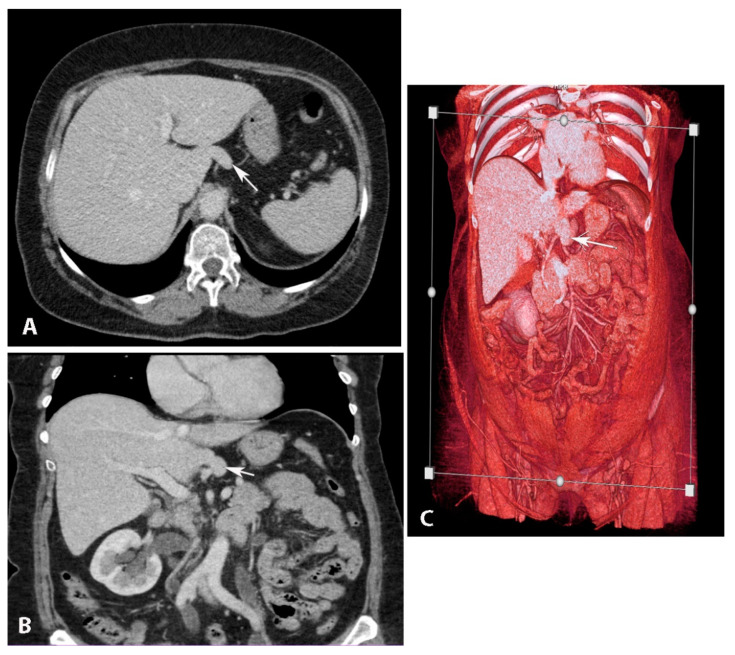
Elongated papillary process (author’s own material). (**A**) Transverse abdominal CT at the level of the liver with elongated papillary process present. (**B**) Coronal abdominal CT where papillary liver process projection can be seen just above the portal vein. (**C**) CT Volume rendered image of the abdomen showing the papillary liver process in its typical location. Arrows show elongated papillary process.

**Figure 9 diagnostics-13-02371-f009:**
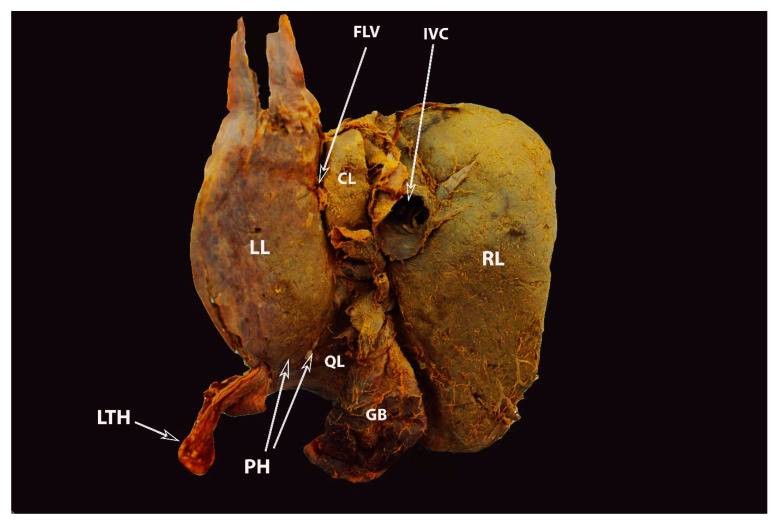
Complete pons hepatis—type III (posterior liver surface). The fissure for ligamentum teres is absent (own material). LL—left liver lobe; RL—right liver lobe; FLV—fissure for ligamentum venosum; IVC—inferior vena cava; QL—quadrate lobe; CL—caudate lobe; GB—gall bladder; LTH—ligamentum teres hepatis; PH—pons hepatis.

**Figure 10 diagnostics-13-02371-f010:**
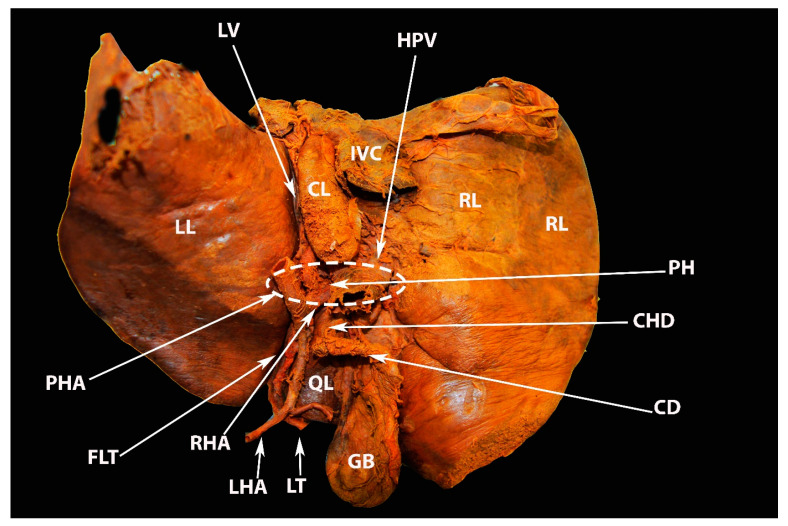
Ligaments on the posterior (visceral) surface of the liver and porta hepatis (author’s own material). LL—left liver lobe; RL—right liver lobe; CL—caudate lobe; QL—quadrate lobe; GB—gall bladder; IVC—inferior vena cava; FLT—fissure for ligamentum teres; LT—ligamentum teres; LHA—left hepatic artery; RHA—right hepatic artery; PHA—proper hepatic artery; HPV—hepatic portal vein; CD—cystic duct; CHD—common hepatic duct; PH—porta hepatis.

**Figure 11 diagnostics-13-02371-f011:**
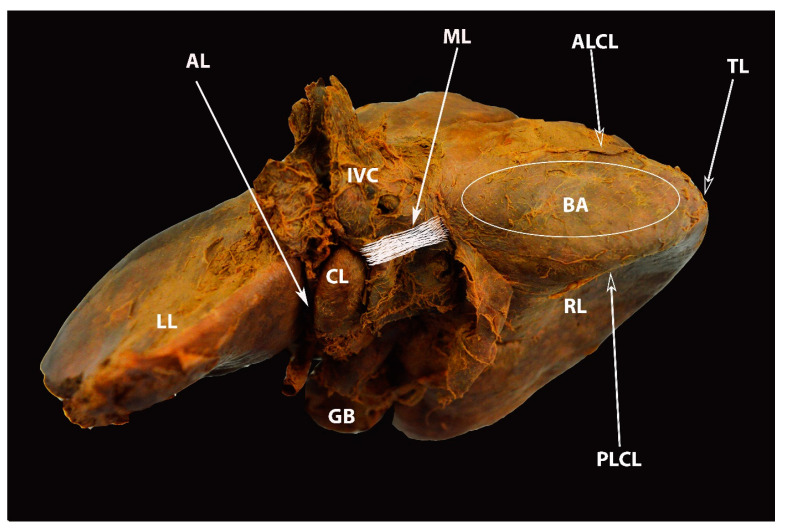
Superior surface of the liver—part of the diaphragmatic surface (author’s own material). LL—left liver lobe; RL—right liver lobe; CL—caudate lobe; GB—gall bladder; AL—Arantius’ ligament; ML—Makuuchi ligament; ALVL—anterior layer of the coronary ligament on the right liver lobe; PLCL—posterior layer of the coronary ligament on the right liver lobe; TL—right triangular ligament; BA—bare area.

**Figure 12 diagnostics-13-02371-f012:**
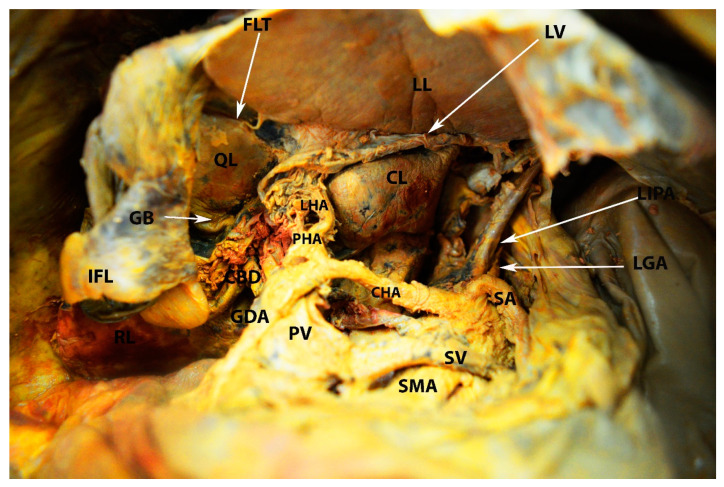
Liver ligament and their relationship with the lesser sac (omental bursa) (the author’s own material)—inferior and posterior surface of the liver. RL—right liver lobe; LL—left liver lobe; CL—caudal lobe; QL—quadrate lobe; GB—gall bladder; IFL—incised falciform ligament; FLT—fissure for ligamentum teres; LV—ligamentum venosum; LIPA—left inferior phrenic artery; LGA—left gastric artery; SA—splenic artery; CHA—common hepatic artery; SV—splenic vein; SMA—superior mesenteric artery; GDA—gastroduodenal artery; PV—portal vein; PHA—proper hepatic artery; LHA—left hepatic artery; CBD—common bile duct.

**Figure 13 diagnostics-13-02371-f013:**
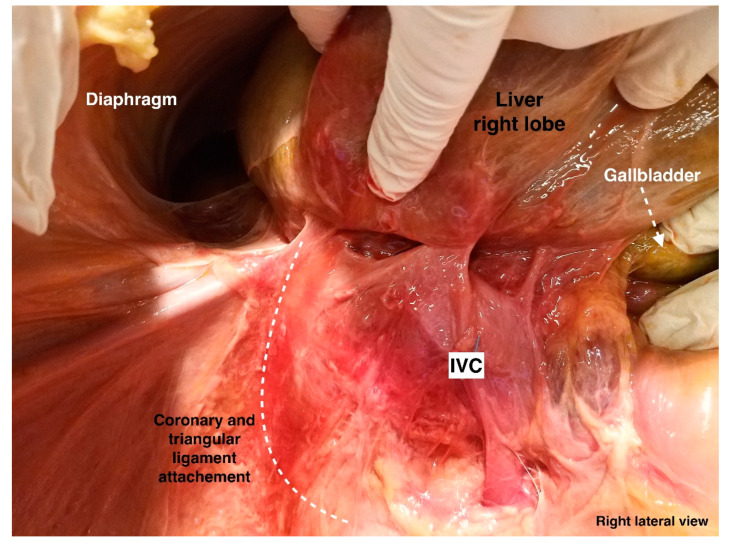
Attachments of coronary ligament and right triangular ligament (from the cadaveric dissection—archive of Dr. Ilker Selcuk). IVC—inferior vena cava.

**Figure 14 diagnostics-13-02371-f014:**
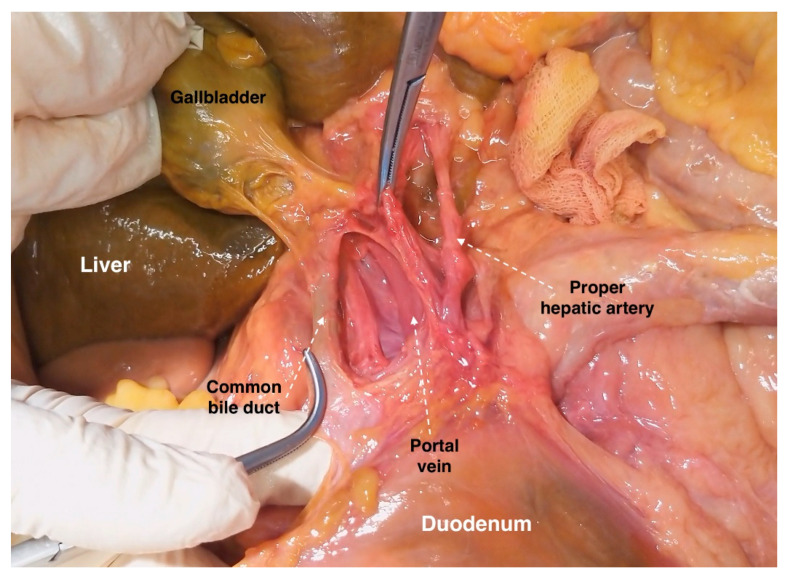
Anatomy of the gastroduodenal ligament (from the cadaveric dissection—archive of Dr. Ilker Selcuk).

**Figure 15 diagnostics-13-02371-f015:**
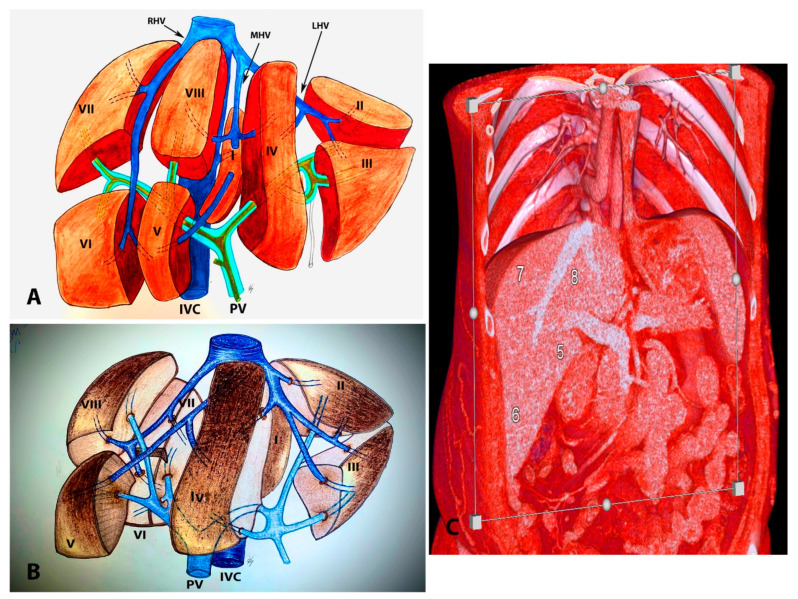
Segments of the liver according to Couinaud’s classification (own material): (**A**) ex vivo appearance; (**B**) in vivo appearance; (**C**) coronal volume-rendered abdominal CT image with frontal plane cut showing the annotated right hepatic lobe segment. Arabic and Latin numbers represents liver segment according to Couinaud’s classification. PV – portal vein; IVC – inferior vena cava.

**Figure 16 diagnostics-13-02371-f016:**
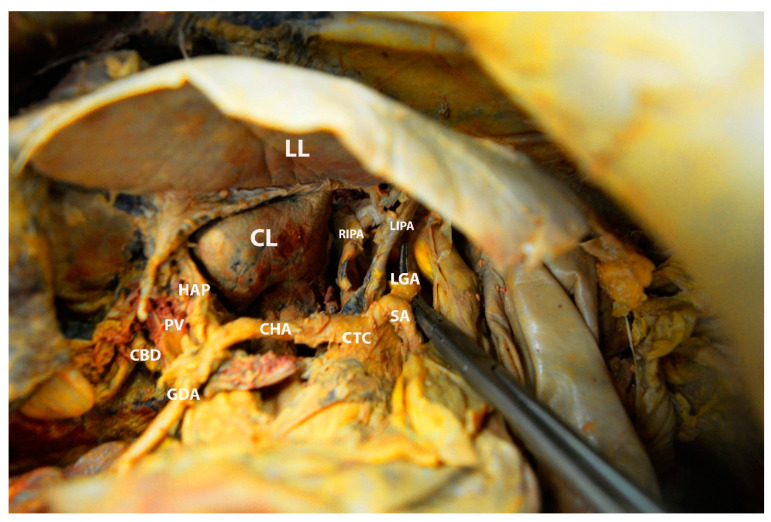
Truncus celiacus and arterial supply of the liver (the author’s own material)—inferior and posterior surface of the liver. CL—caudate lobe; LL—left lobe; CTC—celiac trunk; SA—splenic artery; LGA—left gastric artery; LIPA—left inferior phrenic artery; RIPA—right inferior phrenic artery; CHA—common hepatic artery; GDA—gastroduodenal artery; HAP—proper hepatic artery; PV—portal vein; CBD—common bile duct.

**Figure 17 diagnostics-13-02371-f017:**
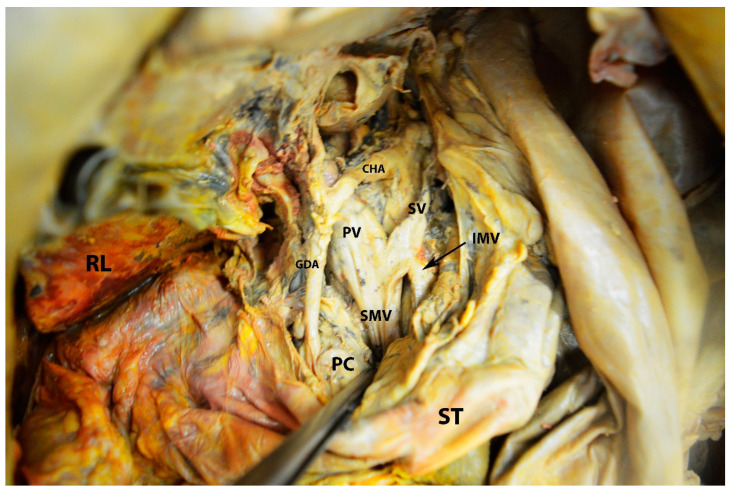
Gross anatomy of the portal vein. The stomach and pancreas are retracted caudally to better visualize the tributaries of the portal vein (author’s own material). RL—right lobe of liver; ST—stomach; PC—pancreas; CHA—common hepatic artery; GDA—gastroduodenal artery; PV—portal vein; SMV—superior mesenteric vein; IMV—inferior mesenteric vein; SV—splenic vein.

**Figure 18 diagnostics-13-02371-f018:**
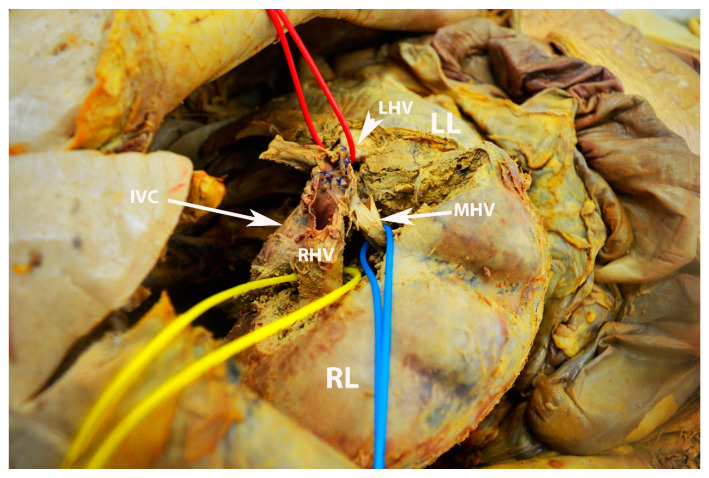
Gross anatomy of hepatic veins—superior surface of the liver (author’s own material). LL—left liver lobe; RL—right liver lobe; IVC—inferior vena cava; LHV—left hepatic vein; MHV—middle hepatic vein; RHV—right hepatic vein.

**Figure 19 diagnostics-13-02371-f019:**
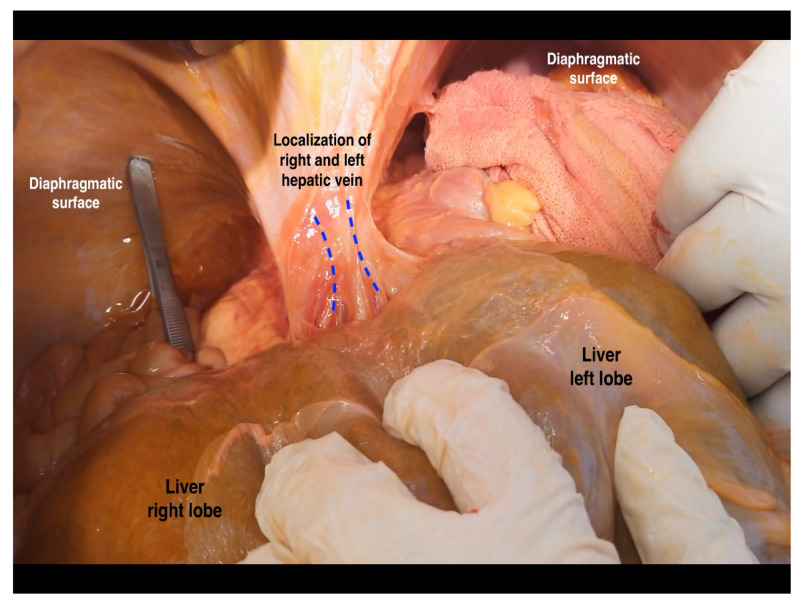
Right and left hepatic vein (from the cadaveric dissection—archive of Dr. Ilker Selcuk).

**Figure 20 diagnostics-13-02371-f020:**
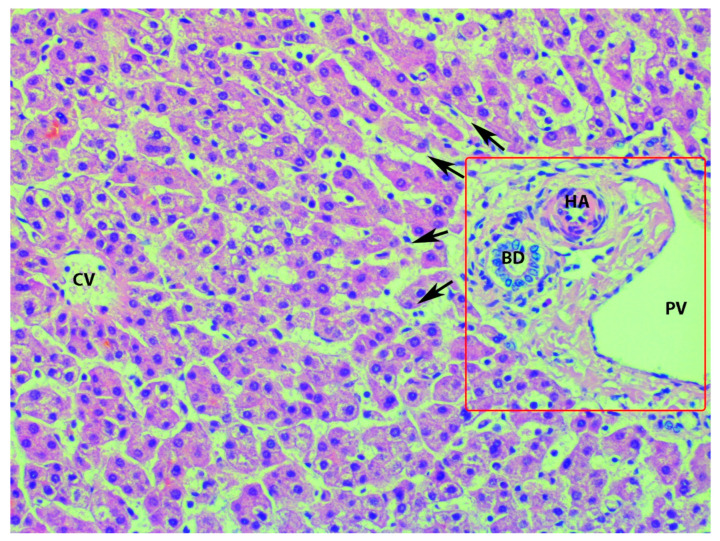
Microanatomy of the liver lobule (author’s own material). PV—branch of portal vein; BD—branch of the bile duct; HA—branch of hepatic artery; CV—central vein; arrows—sinusoids; red square—the portal triad.

**Figure 21 diagnostics-13-02371-f021:**
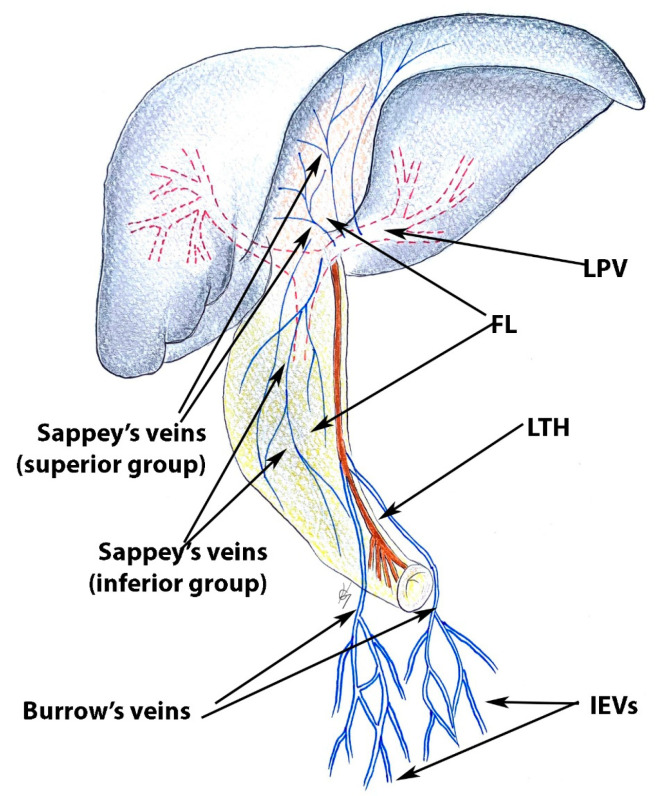
Sappey’s and Burrow’s veins ( adapted with permission from reference [63]). LTH—ligamentum teres hepatis; FL—falciform ligament; LPV—left portal vein; IEVs—inferior epigastric veins.

**Figure 22 diagnostics-13-02371-f022:**
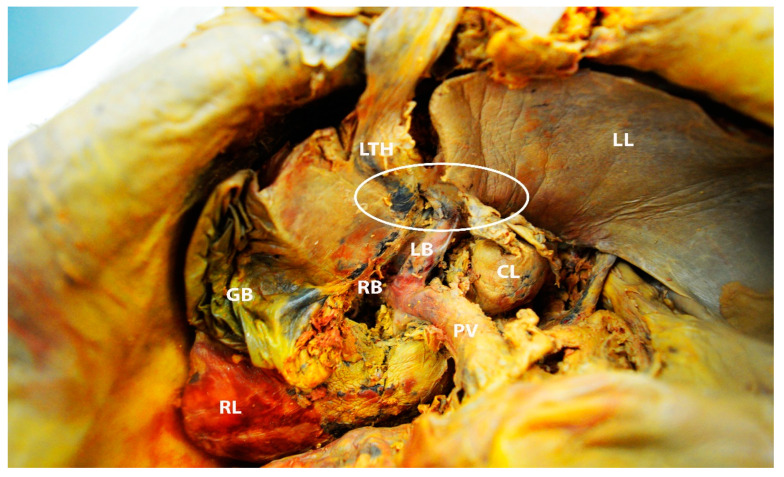
The white ellipse shows the narrow distance between the base of the ligamentum teres hepatis and the left branch of the portal vein (author’s own material). LTH—ligamentum teres hepatis; LL—left liver lobe; RL—right liver lobe; CL—caudate lobe; GB—gallbladder; PV—portal vein; RB—right branch of the portal vein; LB—left branch of the portal vein.

**Figure 23 diagnostics-13-02371-f023:**
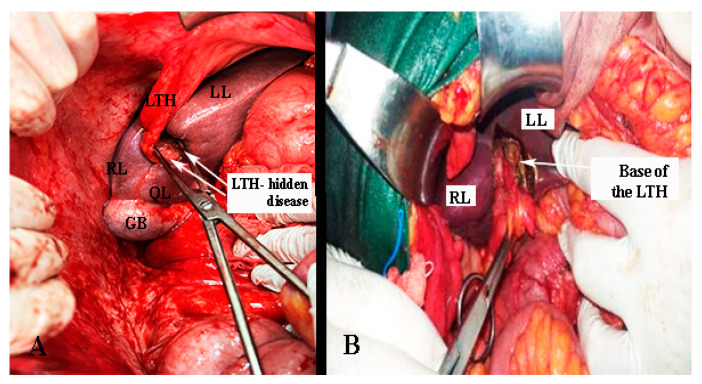
Base of the ligamentum teres hepatis (author’s own material—open surgery). (**A**) Peritoneal tumor dissemination of the base of the ligamentum teres hepatis in case of advanced ovarian cancer. (**B**) Removal of macroscopic peritoneal metastases at the base of the ligament. The round ligament of the liver was entirely removed. LTH—ligamentum teres hepatis, LL—left lobe of the liver; RL—right lobe of the liver; QL—quadrate lobe; LTH—ligamentum teres hepatis, GB—gall bladder.

**Figure 24 diagnostics-13-02371-f024:**
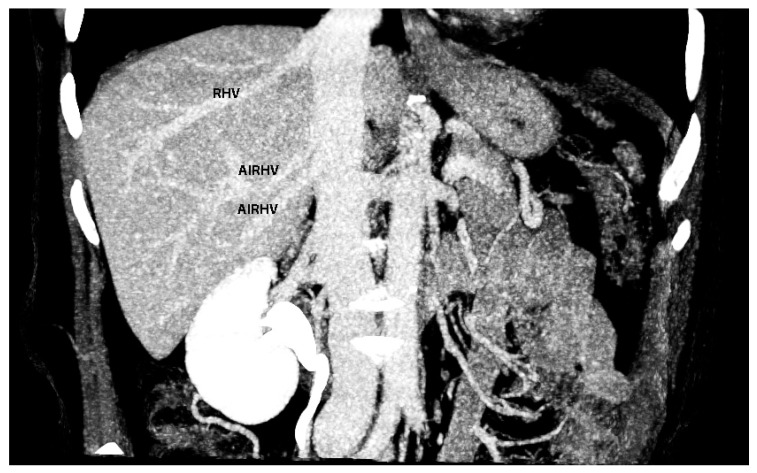
Coronal post-contrast CT image of the abdomen, showing two accessory right inferior hepatic veins of the liver (author’s own material). RHV—right hepatic vein; AIRHV—accessory inferior right hepatic vein.

**Figure 25 diagnostics-13-02371-f025:**
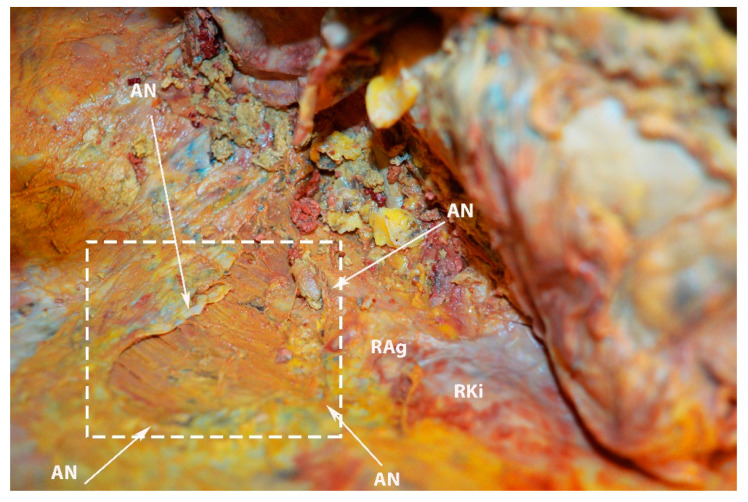
Limits of the bare area of the liver on the diaphragmatic surface (own material). (The liver was removed for better identification.) AN—area nuda; RAg—right adrenal gland; RKi—right kidney.

**Figure 26 diagnostics-13-02371-f026:**
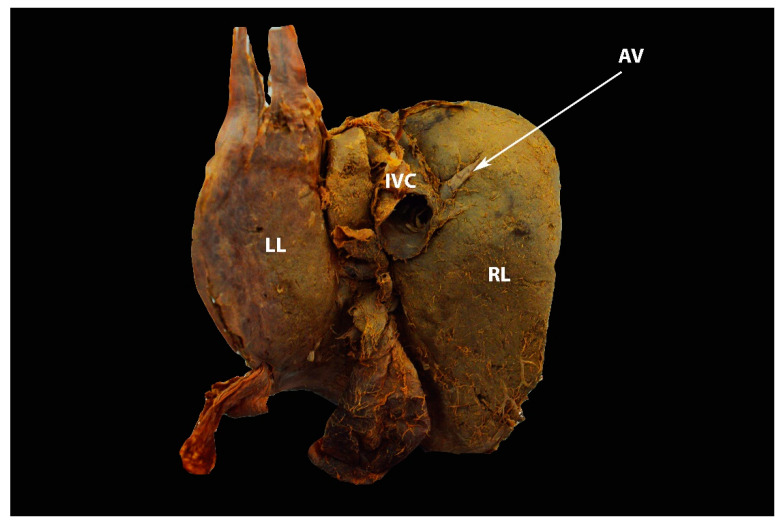
Small retrohepatic additional right liver lobe vein, which drains directly into the IVC—visceral surface of the liver (author’s own material). IVC—inferior vena cava; AV— additional vein; LL—left liver lobe; RL—right liver lobe.

**Figure 27 diagnostics-13-02371-f027:**
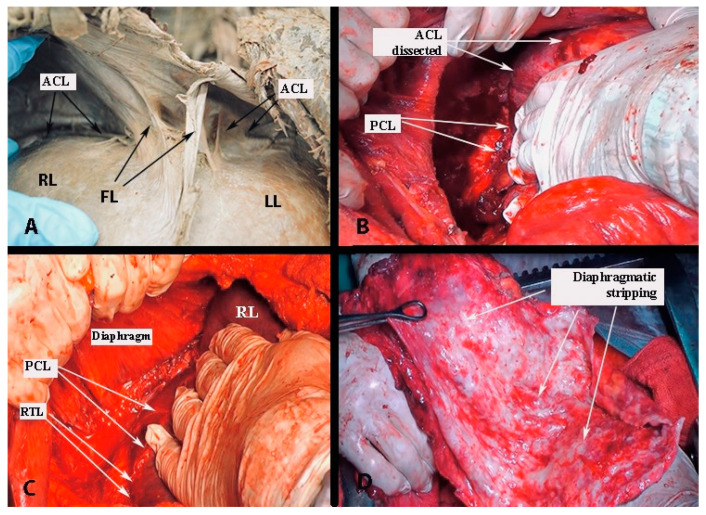
Some steps of liver mobilization followed by diaphragmatic stripping (author’s own material). (**A**) Step 1—dissection of falciform ligament (embalmed cadaver). (**B**,**C**) Steps 6,7, 8—dissection of posterior layer of the coronary ligament on the right liver lobe and dissection of the right triangular ligament. (**C**) Final view after diaphragmatic stripping in case of massive metastases on the right diaphragmatic peritoneum ((**B**–**D**) open surgery). ACL—anterior layer of the coronary ligament; RL—right lobe of the liver; LL—left lobe of the liver; FL—falciform ligament; PCL—posterior layer of the coronary ligament on the right liver lobe; RTL—right triangular ligament.

**Figure 28 diagnostics-13-02371-f028:**
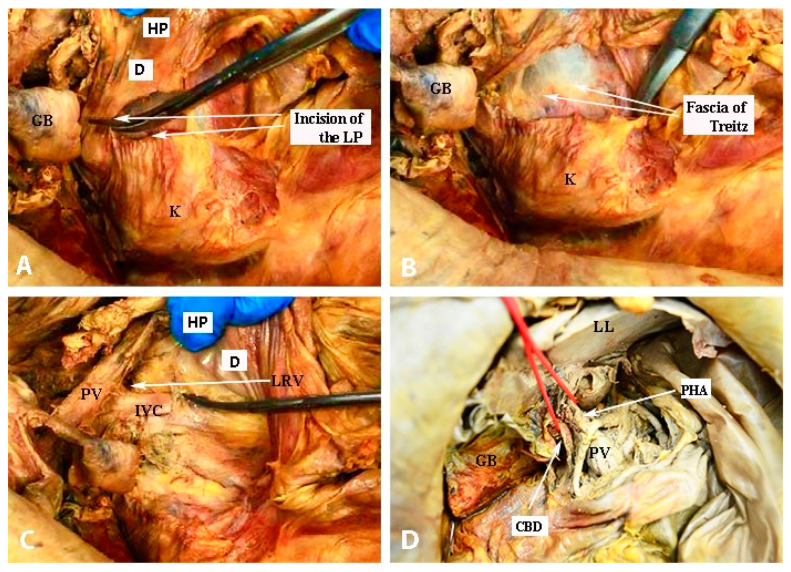
Kocher (**A**–**C**) and Pringle (**D**) maneuvers (embalmed cadavers—author’s own material). **(A)** Incision of the lateral peritoneum, which started between the lateral aspect of the epiploic foramen and the inferior duodenal flexure. (**B**) Dissection of fascia of Treitz, which is located below the head of the pancreas. (**C**) Exposure of the inferior vena cava until the medial limit of dissection—the left renal vein. (**D**) Vessel loop, which encircles the hepatoduodenal ligament. HP—head of the pancreas, D—duodenum, GB—gall bladder; K— kidney; PV—portal vein, LRV—left renal vein; prop IVC—inferior vena cava; PHA—proper hepatic artery; CBD—common bile duct; LL—left lobe of the liver.

## Data Availability

Authors declare that all related data are available concerning researchers by the corresponding author’s email.

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
