# Peer review of "Surgical Anatomy of the Liver—Significance in Ovarian Cancer Surgery"

_diagnostics, 2023, doi:10.3390/diagnostics13142371_

Round 1

Reviewer 1 Report

In this narrative review, Kostov and colleagues described liver anatomy for ovarian metastasis surgery. They provided a well-presented revision of the liver's gross appearance and anatomical variations that can potentially mislead the proper identification of ovarian cancer metastasis. Representative images and schemes are also provided.

Overall, the review is interesting and provides a well-documented description of liver anatomy for surgical reference.

Please, consider the following comments:

- Page 5, lines 143-148: explain better the potential association between diaphragmatic grooves and the Chilaiditi sign. 

- Page 7, lines 164-166: explain better the question here presented (why is it unanswered? What are the issue related to metastasis located in this lobe, if any?), document the references that make the Authors report this question, and provide the Authors' opinion about this topic. 

- Page 18, lines 436: references are required, indeed. 

Author Response

In this narrative review, Kostov and colleagues described liver anatomy for ovarian metastasis surgery. They provided a well-presented revision of the liver's gross appearance and anatomical variations that can potentially mislead the proper identification of ovarian cancer metastasis. Representative images and schemes are also provided.

Overall, the review is interesting and provides a well-documented description of liver anatomy for surgical reference.

Please, consider the following comments:

- Page 5, lines 143-148: explain better the potential association between diaphragmatic grooves and the Chilaiditi sign.

Author’s Reply:

We agree with the reviewer. It is done as recommended. 

The next text was inserted :

Diaphragmatic grooves could be mistaken with Chilaiditi sing or syndrome, especially in cases of free air due to a perforation of abdominal organ [22].

Chilaiditi sign (CS) is defined as interposition of the colon (commonly transverse mesocolon), between the right liver lobe and the diaphragm. The sign represents a radiological finding of a gas between the right diaphragm and the right lobe of the liver [23-27].

The CS is known as Chilaiditi syndrome when it is accompanied by symptoms ( pain, vomiting, constipation)  and complications (intestinal obstruction, perforation and ischemia) [22-27]. There are many theories for this untypical predisposition of the bowel – diaphragmatic ( phrenic nerve palsy or congenital muscle loss), hepatic ( weakness of the falciform ligament), abnormally long colon, ascites and obesity [22,25-27].  There is also a theory, which states that the diaphragmatic grooves are formed by a mesocolic tissue invasion of the adjacent anterior right liver lobe margins [23]. Yavuz et al. noticed the possible relation between diaphragmatic grooves, CS and Chilaiditi syndrome. Therefore, authors retrospectively investigated this possible connection on 2314 CT scans. Authors did not found statistical or significant correlation between diaphragmatic grooves and the syndrome. However, authors concluded that the grooves are likely derived from the CS, as more than half of the patients with CS had diaphragmatic grooves (25 patients (54.3%) of 46 had grooves on the right liver lobe near the falciform ligament). Their theory is supported by the fact that the majority of grooves are found among adult population. Nevertheless, Yavuz et al. mentioned that further studies are needed [23].

It should be stressed that Chilaiditi syndrome could be confused radiologically with diaphragmatic grooves. Cawich et al. reported for a case of a patient with peptic ulcer perforation at the first part of the duodenum. The authors initially considered possible Chilaiditi syndrome as the patient had an air above the right lobe of the liver on preoperative radiographs.  Intraoperative findings showed diaphragmatic grooves on the right liver lobe. Authors concluded that a true pneumoperitoneum with the presence of diaphragmatic grooves could be mistaken with Chilaiditi syndrome [22].

The next references were inserted:

22. Cawich, S. O., Spence, R., Mohammed, F., Gardner, M. T., Sinanan, A., & Naraynsingh, V. (2017). The liver and Chilaiditi's syndrome: Significance of hepatic surface grooves. SAGE open medical case reports, 5, 2050313X17744979. https://doi.org/10.1177/2050313X17744979

  1. Gulati MS, Wafula J, Aggarwal S: Chilaiditi’s sign possibly associated with malposition of chest tube placement. J Postgrad Med. 2008, 54:138-139. 10.4103/0022-3859.40781
  2. Moaven O, Hodin RA: Chilaiditi syndrome: a rare entity with important differential diagnoses. Gastroenterol Hepatol (N Y). 2012, 8:276-278.
  3. Sohal, R. J., Adams, S. H., Phogat, V., Durer, C., & Harish, A. (2019). Chilaiditi's Sign: A Case Report. Cureus, 11(11), e6230. https://doi.org/10.7759/cureus.6230

- Page 7, lines 164-166: explain better the question here presented (why is it unanswered? What are the issue related to metastasis located in this lobe, if any?), document the references that make the Authors report this question, and provide the Authors' opinion about this topic.

 Author’s Reply:

 We agree with the reviewer. There is no reference. We asked that question, as it is anomaly, which we believe has been neglected by oncogynecologist over the years, as it is not so rare!   Mainly general surgeons are familiar with Riedel’s lobe! As there are no reference, we removed the question, but further discussed the topic below with appropriate citations.

 The next text was inserted :

 A few cases of primary malignant tumors or metastases to the Riedel’s lobe have been described. [32-34].  Soo et al. reported for a Riedel’s lobe metastasis from a ductal breast cancer [32]. Zamfir et al. observed a case of a 65-years-old woman with primary hepatocellular carcinoma arising from the Riedel’s lobe.  The lobe was resected with "en-block" cholecystectomy [33]. Handola et al. reported a case of 64-year-old woman with incidental observation of Riedel’s lobe and intrahepatic cholangiocarcinoma. Authors stated that there are unanswered association between Riedel’s lobe and cancer. They concluded that the lobe could be considered a possible site for primary hepatocellular carcinoma or hidden metastases [34]. Notably, the majority of cases of Riedel’s lobe involvement by a malignant tumor affected female population [32-34]. However, there is no reported case in medical literature of metastases to the Riedel’s lobe by ovarian cancer. Perhaps, there were such cases, but this liver pathology was probably neglected by oncogynecologists. Moreover, some authors believe that the lobe is a simple variant of liver anatomy, corresponding to hypertrophy of segments V and VI, rather than a true anatomical variation [35-37]. Additionally, the lobe can be a source of a living-related” hepatic transplant [13, 28].   Therefore, ovarian cancer metastases to the Riedel’s lobe should be staged as FIGO stage IV, as this liver anomaly is actually part of the liver.

The next references was incorporated:

  1. SOO, M. S. C., & ADATEPE, M. H. (1990). Metastatic Lesions Arising in a Riedel??s Lobe: Findings from a Sulfur Colloid Liver-Spleen Scan. Clinical Nuclear Medicine, 15(11), 814–815. doi:10.1097/00003072-199011000-00010
  1. amfir, R., BraÅŸoveanu, V., BoroÅŸ, M., Herlea, V., & Popescu, I. (2008). Hepatocellular carcinoma in Riedel's lobe. Chirurgia (Bucharest, Romania : 1990), 103(1), 121–123.
  1. Al-Handola R, Chinnappan J, Bakeer M, et al. (June 20, 2023) Incidental Finding of Riedel’s Lobe of the Liver and Intrahepatic Cholangiocarcinoma. Cureus 15(6): e40683. doi:10.7759/cureus.40683
  1. KUDO, M. (2000). Riedel’’s Lobe of the Liver and Its Clinical Implication. Internal Medicine, 39(2), 87–88. doi:10.2169/internalmedicine.39.87
  1. Lane D (1966) Masses in the right hypochondrium and Riedel's lobe. Med J Aust 1:896–899.
  1. De Simoni O, Barina A, Gruppo M, Scapinello A, Mourmouras V, Pilati P, Franzato B. A Potentially Misleading Hepatocellular Carcinoma. Medicina. 2021; 57(8):850. https://doi.org/10.3390/medicina57080850

- Page 18, lines 436: references are required, indeed.

Author’s reply:

The references were incorporated.

We are grateful for your valuable time and effort in reviewing our manuscript.

Based on your useful and scientific comments, we believe our manuscript has been improved to a higher level.

=========================================================

Reviewer 2 Report

This is one of the most interesting manuscripts I have reviewed. Congratulations for the good work

The following is suggested:

1- Adding a combined image of real operative photos for the steps of liver mobilization

2- Adding a paragraph on tips of porta hepatis and gall bladder fossa dissection in case of being affected by the tumor spread

3- some typing errors are noted

Author Response

This is one of the most interesting manuscripts I have reviewed. Congratulations for the good work

The following is suggested:

1- Adding a combined image of real operative photos for the steps of liver mobilization

Author’s Reply: We agree with the reviewer. We incorporated some of the steps during the procedure, as initial steps are very hard to be recorded, as the dissection is just below the sternum and ribs. Moreover, the quality of these photos is not good as they were took from video files! We try to choose the best quality figures at the end of the mobilization and during the final steps of the procedure. We incorporated two figures – 23 and 27.

2- Adding a paragraph on tips of porta hepatis and gall bladder fossa dissection in case of being affected by the tumor spread

Author’s Reply: We agree with the reviewer. For the bladder fossa we only explained dissection of the peritoneum as we do not feel quite experience to comment cholecystectomy. Moreover, as mentioned above gall-bladder is not the topic of the article.

The next paragraphs were incorporated in the text:

VIII.2. Porta hepatis and hepatoduodenal ligament dissection

The dissemination of porta hepatis from ovarian cancer varies in medical literature, as authors report for both – peritoneal and lymphatic tumor spread [96 -98]. Raspagliesi found that 19 % of patients with advanced ovarian cancer had peritoneum dissemination at porta hepatis [96]. Tozzi et al. reported that 14.3% of examined patients with advanced ovarian cancer had dissemination of porta hepatis and hepato-celiac lymph nodes. Eighteen patients out of 31 had only porta hepatis peritoneal involvement [97]. Donato et al. reported for 4.5% portal nodes metastases among 55 women with advanced ovarian cancer and hepatobiliary involvement [98].

There are different techniques for porta hepatis dissection. Some authors use vessel loop through the epiploic foramen to encircle the hepatoduodenal ligament (part of Pringle maneuver), whereas others carry out the Kocher maneuver to provide enough space for dissection. Tozzi et al. performs both maneuvers during dissection. The Kocher maneuver represents a medial pancreato-duodenal mobilization. The head of the pancreas and the first, second and proximal third portions of the duodenum are mobilized. The procedure is performed easily as there is an avascular plane below the duodenum and the pancreatic head. The mobilization continues at the level, where the left renal vein drains into the IVC (figure 28 ) [96-99].

The Pringle maneuver represents a vessel loop, which encircles the hepatoduodenal ligament and its structure (figure 28). It is used to minimize blood loss during different types of hepatic resection [100].

The dissection starts with opening the anterior peritoneum of the hepatoduodenal ligament at a tumor free area. The proper hepatic artery and the common bile duct are identified. The vessels loop is retracted medially and the posterior peritoneum of the ligament is dissected from the dorsal aspect of the portal vein. When all three anatomical structures of the hepatoduodenal ligament are identified and mobilized, the dissection continues in a retrograde fashion until the hepatic hilum. Enlarged lymph nodes are resected. Moreover, the peritoneal stripping of the ligament continues medio-laterally and posteriorly to complete the circumferential dissection. Surgeons should be aware of vessels variations (replaced right hepatic artery or left hepatic artery arising from the superior mesenteric artery) or biliary tree variations [96-100, 102].

VIII.3. Gall-bladder fossa dissection

Ovarian tumor dissemination to the gall-bladder fossa is often followed by a cholecystectomy. Rosati et al. mentioned that in such cases a hepatobiliary surgeon is required [7]. The peritoneal tissue covering the gall-bladder is a common anatomical area of dissemination. In such cases, surgeons dissect the peritoneal fold between the gall-bladder and the duodenum in order to identify the disease. Gall-bladder removal should be performed only in cases when optimal cytoreduction could be achieved [3, 102].

One additional figure was incorporated – Figure 28

The next references was inserted:

  1. Raspagliesi, F., Ditto, A., Martinelli, F., Haeusler, E., & Lorusso, D. (2013). Advanced ovarian cancer: omental bursa, lesser omentum, celiac, portal and triad nodes spread as cause of inaccurate evaluation of residual tumor. Gynecologic oncology, 129(1), 92–96. https://doi.org/10.1016/j.ygyno.2013.01.024
  2. Tozzi, R., Traill, Z., Garruto Campanile, R., Ferrari, F., Soleymani Majd, H., Nieuwstad, J., Hardern, K., & Gubbala, K. (2016). Porta hepatis peritonectomy and hepato-celiac lymphadenectomy in patients with stage IIIC-IV ovarian cancer: Diagnostic pathway, surgical technique and outcomes. Gynecologic oncology, 143(1), 35–39. https://doi.org/10.1016/j.ygyno.2016.08.232
  3. Di Donato, V., Giannini, A., D'Oria, O., Schiavi, M. C., Di Pinto, A., Fischetti, M., Lecce, F., Perniola, G., Battaglia, F., Berloco, P., Muzii, L., & Benedetti Panici, P. (2021). Hepatobiliary Disease Resection in Patients with Advanced Epithelial Ovarian Cancer: Prognostic Role and Optimal Cytoreduction. Annals of surgical oncology, 28(1), 222–230. https://doi.org/10.1245/s10434-020-08989-3
  4. Livani, A., Angelis, S., Skandalakis, P. N., & Filippou, D. (2022). The Story Retold: The Kocher Manoeuvre. Cureus, 14(9), e29409. https://doi.org/10.7759/cureus.29409
  5. Son, J. H., & Chang, S. J. (2021). Cholecystectomy, porta hepatis stripping, and omental bursectomy. Gland surgery, 10(3), 1230–1234. https://doi.org/10.21037/gs-2019-ursoc-02
  6. Mownah, O. A., & Aroori, S. (2023). The Pringle maneuver in the modern era: A review of techniques for hepatic inflow occlusion in minimally invasive liver resection. Annals of hepato-biliary-pancreatic surgery, 27(2), 131–140. https://doi.org/10.14701/ahbps.22-109
  7. Bhandoria G, Bhatt A, Mehta S, Glehen O. Upper-Abdominal Cytoreduction for Advanced Ovarian Cancer—Therapeutic Rationale, Surgical Anatomy and Techniques of Cytoreduction. Surgical Techniques Development. 2023; 12(1):1-33. https://doi.org/10.3390/std12010001

3- some typing errors are noted

Author’s Reply: We tried to correct some of the errors.

We are grateful for your valuable time and effort in reviewing our manuscript.

Based on your useful and scientific comments, we believe our manuscript has been improved to a higher level.